# Clinical practice guideline recommendations to improve the mental health of adult trauma patients: protocol for a systematic review

Mélanie Bérubé [ID],[1,2] Nori Bradley,[3] Meaghan O'Donnell,[4] Henry Thomas Stelfox,[5] Naisan Garraway,[6] Helen-Maria Vasiliadis,[7] Valerie Turcotte,[8] Michel Perreault,[9] Matthew Menear,[10] Léonie Archambault,[9] Juanita Haagsma,[11] Hélène Provencher,[2] Christine Genest,[12,13] Marc-Aurèle Gagnon,[1] Laurence Bourque,[1] Alexandra Lapierre [ID],[1] Amal Khalfi,[2] William Panenka[14]

For numbered affiliations see end of article.

**Correspondence to**
Mélanie Bérubé;
melanie.berube@fsi.ulaval.ca

## ABSTRACT

**Introduction** Mental disorders are common in adult patients with traumatic injuries. To limit the burden of poor psychological well-being in this population, recognised authorities have issued recommendations through clinical practice guidelines (CPGs). However, the uptake of evidence-based recommendations to improve the mental health of trauma patients has been low until recently. This may be explained by the complexity of optimising mental health practices and interpretating CGPs scope and quality. Our aim is to systematically review CPG mental health recommendations in the context of trauma care and appraise their quality.

**Methods and analysis** We will identify CPG through a search strategy applied to Medline, Embase, CINAHL, PsycINFO and Web of Science databases, as well as guidelines repositories and websites of trauma associations. We will target CPGs on adult and acute trauma populations including at least one recommendation on any prevention, screening, assessment, intervention, patient and family engagement, referral or follow-up procedure related to mental health endorsed by recognised organisations in high-income countries. No language limitations will be applied, and we will limit the search to the last 15 years. Pairs of reviewers will independently screen titles, abstracts, full texts, and carry out data extraction and quality assessment of CPGs using the Appraisal of Guidelines Research and Evaluation (AGREE) II. We will synthesise the evidence on recommendations for CPGs rated as moderate or high quality using a matrix based on the Grading of Recommendations Assessment, Development and Evaluation quality of evidence, strength of recommendation, health and social determinants and whether recommendations were made using a population-based approach.

**Ethics and dissemination** Ethics approval is not required, as we will conduct secondary analysis of published data. The results will be disseminated in a peer-reviewed journal, at international and national scientific meetings. Accessible summary will be distributed to interested parties through professional, healthcare quality and persons with lived experience associations.

---

## STRENGTHS AND LIMITATIONS OF THIS STUDY

⇒ We will produce a meta-synthesis of clinical practice guidelines (CPGs) recommendations on mental health in injury care based on their methodological quality and level of evidence.

⇒ Our review involves interested parties to maximise its relevance and reach.

⇒ We will focus solely on CPGs for mental health from high-income countries.

⇒ Our search strategy is not designed to identify CPGs that do not specifically focus on mental health in trauma populations.

---

**PROSPERO registration number** (ID454728).

## INTRODUCTION

Improvements in trauma care in high-income countries over the past 30 years have reduced mortality from as high as 50%–11%.[1 2] As a result, more and more patients survive their traumatic injuries, but still face a significant burden of morbidity. Indeed, traumatic injuries are a leading cause of disability,[3 4] and result in the highest number of productive years of life lost.[5 6]

The impact of traumatic injuries is not without consequences for patients' mental health, not to mention the fact that many are already vulnerable on a psychosocial level. Before the injury, 20%–40% of trauma patients have mental health issues, including substance use disorders and mood disorders,[7 8] which can impair recovery.[9] After the injury, up to 45% of patients develop acute stress disorder during their hospital stay,[10] and 10%–50% post-traumatic stress disorder in many high-income countries, with prevalence varying according to trauma mechanisms and

recovery phases.[9 11–19] Hence, patients who sustained traumatic injuries have a 40% higher risk of being hospitalised for a mental disorder when comparing the 5 years before and after the injury.[19] Cumulative structural, social and health determinants have been shown to contribute significantly to mental disorders among trauma patients with racial/ethnic minorities, those with low levels of education and financial resources, males or patients identifying themselves as men and those living in rural areas being most vulnerable.[19 20]

The high prevalence of mental disorders in trauma patients is alarming considering that they have profound negative impacts on individual and social outcomes. Trauma patients suffering from mental disorders were shown to have twice as many complications and twice the length of hospital stays.[21] Similarly, these patients are almost 10 times more likely to develop chronic pain and have physical limitations, and two to four times more likely not to return to work than patients without mental disorders,[22] while having a poorer quality of life.[14]

Given the growing body of research evidence highlighting the burden of mental health issues following traumatic injury, recognised authorities[23–26] have proposed clinical practice guidelines (CPGs) to reduce their prevalence. CPG recommendations apply to all healthcare professionals (HCPs) involved in the trauma care continuum, and cover mental health promotion/illness prevention, screening, brief and more comprehensive interventions, patient referral to subsequent specialty services and follow-up, and patient and family centered-care approach (ie, collaborative approach, share-decision making[27]). HCPs' adherence rates to CPGs just prior to the publication of key CPGs were issued in late 2022, such as the American College of Surgeons-Committee on Trauma on Screening and Intervention for Substance Use in the Acute Trauma Patients,[23] were are reported to be less than 30%.[28 29] While uptake rates may have risen since then, the fact remains that improving mental health practices in an area where HCPs have more expertise in physical health[19 30] can be complex and put pressure on increasingly limited resources of healthcare systems. It is therefore crucial to summarise the recommendations on the prevention and treatment of mental disorders in the context of trauma, and identified those with the greatest support for implementation, to enable organisations in different countries to target the priorities on which to focus. Hence, we aim to systematically review CPG recommendations for mental health in the context of injury care, and to appraise their quality.

## METHODS

In line with the knowledge synthesis phase of the Knowledge to Action framework,[31] we will conduct a knowledge synthesis on CPGs. This protocol was developed according to Cochrane recommendations for systematic reviews (SR)[32] and methodological guidelines for SR on CPGs[33] and is reported according to the Preferred Reporting Items for Systematic review and Meta-Analysis Protocol statement.[34]

### Patient and public involvement

To maximise contributions, we set up an advisory committee including representatives of associations that play a role in setting standards for trauma and mental healthcare (Trauma Association of Canada, the Mental Health Commission of Canada, and the *Institut national d'excellence en santé et en services sociaux*) and in supporting persons with lived experience (Brain Injury Canada, Mothers Against Drunk Driving—Canada, *Moelle épinière et motricité Québec and Connexion TCC-Québec*). The advisory committee also includes HCPs from various disciplines involved with trauma patients (specialised physicians/surgeons, nurses, rehabilitation professionals, psychologists and social workers), persons with lived experience and decision makers. The committee was involved in the development of the protocol and will oversee the progress of the review and take part in disseminating the results.

### Eligibility

We will target CPGs on adult (≥18 years old) and in-patient acute trauma populations (P) including at least one recommendation (R) on any prevention, screening, assessment, intervention, patient and family engagement, referral or follow-up procedure related to mental health (I) with any or no comparator (C) endorsed by recognised organisations in high-income countries within the last 15 years (September 2008 to a maximum of 6 months prior to submission) (A). No language restriction will be applied. We will use an open-access online translator (https://www.deepl.com/translator) for studies not written in English of French.[35] CGPs are defined as 'statements that include recommendations intended to optimise patient care that are informed by a review of evidence and an assessment of benefits and harms of alternative care options'.[36] We will include CPGs focusing on the categories of mental disorders defined in the Diagnostic and Statistical Manual of Mental Disorders fifth edition-text revision (DSM-5-TR)[37] that most often affect trauma patients: trauma-and-stressor-related disorders, anxiety disorders, substance-related and addictive disorders, suicidal behaviour disorder and self-injury.[7 8 19] High-income countries, based on World Bank definitions,[38] will be targeted to ensure that recommendations are compatible with the practices of accredited trauma centres,[39] while the time limit aims to focus on recent recommendations.

### Search strategy

Based on the recommendations of the Peer Review of Electronic Search Strategies,[40] we will establish a systematic search strategy in collaboration with an information specialist, which will be independently reviewed by a second information specialist. We will use a combination of controlled terms (eg, MeSH for MEDLINE and Emtree for Embase) and free vocabulary according to the

**Table 1** Medline (Ovid) (8 August 2023)

| Concepts | Search strategy keywords | Research and no of results |
|---|---|---|
| Trauma/recovery (controlled vocabulary) (free text) | exp 'Wounds and Injuries'/ or exp Brain Hemorrhage, Traumatic/ or Brain Injuries/ or Coma, Post-Head Injury/ or Craniocerebral Trauma/ or Diffuse Axonal Injury/ or exp Fractures, Bone/ or Head Injuries, Closed/ or Head Injuries, Penetrating/ or exp Intracranial Hemorrhage, Traumatic/ or exp Skull Fractures/ | #1 1 007 620 |
| | Fractur* OR Injur* OR TBI OR trauma* | #2 1 430 550 |
| | 1 OR 2 | #3 1 894 323 |
| Mental health (controlled vocabulary) (free text) | exp Mental Disorders/ or exp Mental Health/ or exp Substance-Related Disorders/ | #4 1 501 202 |
| | 'mental health' OR 'mental disorder*' OR 'mental illness' OR 'post-traumatic stress disorder' OR 'post traumatic stress disorder' OR 'PTSD' OR 'Acute stress disorder*' or 'acute stress reaction*' or 'adjustment disorder*' or 'suicidal behavior*' or 'self-injur*' or 'self injur*' or 'self-harm' or 'self harm' or 'depressive disorder*' or 'depression' or 'mood disorder*' or 'anxiety disorder*' or 'trauma and stressor-related disorder*' or 'trauma and stressor related disorder*' or 'Mental distress' or 'psychological distress' OR ((drug* OR substance OR alcohol OR marijuana OR cannabis OR narcotic* OR opiate* OR opioid* OR opium) Adj4 (abus* OR 'use' OR user OR usage OR misus* OR usin* OR utilis* OR depend* OR addict* OR illegal* OR illicit* OR habit* OR withdraw* OR behavi* OR abstinence* OR abstain* OR intoxica* OR addict* OR disorder*)) OR 'drug rehabilitation' OR 'non-prescription drugs' OR 'non-prescription drug' | #5 1 069 467 |
| | 4 OR 5 | #6 2 121 781 |
| Guideline (controlled vocabulary) (free text) | exp Practice Guidelines as Topic/ or exp Practice Guideline/ or exp Guidelines as Topic/ or exp Guideline/ | #7 208 566 |
| | Guide OR Guideline* | #8 676 237 |
| | 7 OR 8 | #9 784 520 |
| Total | 3 AND 6 AND 9 Limit to 2008–2023 | 3882 |

PTSD, post-traumatic stress disorder.

following themes: wounds and injuries, trauma, mental health, mental disorders and guidelines. We will apply the pilot search strategy in Medline (Ovid) and Embase (Ovid) databases. Relevant words found in the title, abstract and text of retrieved studies will be collected. We will then add these words and indexing terms to the search strategy, which will be executed in the Medline (Ovid), Embase (Ovid), CINAHL (EBSCO), PsycINFO (Ovid) and Web of Science (Clarivate) databases. Relevant GPGs will also be searched in CPGs repositories and trauma association websites, a list of which was drawn up with members of the research team who have in-depth knowledge of the innerworkings in this field. Using a preliminary search strategy (from January 2018 to 8 August 2023; Tables 1 and 2), we have identified 3882 citations, including five sentinel CPGs[24–26 41 42] identified a priori, indicating good sensitivity and specificity.

### CPGs selection and data extraction

We will manage citations using EndNote (V.X9.3.3, New York City: Thomson Reuters, 2018). Pairs of reviewers will independently screen titles, abstracts and full-texts, and carry out data extraction. Once the final set of included CPGs has been obtained, all associated methodological supplements will be retrieved before data extraction or quality assessment begins.[33] Data from included CPGs will be extracted with an extraction form, which will be pilot-tested on five items to ensure feasibility and completeness.[32] For each recommendation within CPGs, we will extract information on the characteristics of included CPGs: title and year, country, organisation, target users, population including determinants contributing to disparities in mental healthcare[43 44] (eg, type of trauma, gender, age group, ethnocultural group, geographical location and socioeconomic status), areas of focus, intervention, quality of evidence and strength of recommendations. Since interventions that use a population-based approach, that is, stepped and collaborative care tailored to patients' risks and needs, have been shown to be more effective than standardised clinical interventions in improving the mental health of trauma patients,[45] we will also extract data on whether recommendations provide details on these aspects. If important information is missing or unclear, we will request it by sending up to three emails to the first, second and last authors.

### Quality

Pairs of reviewers with content expertise will independently assess CPGs quality based on the six domains of the AGREE II tool:[46] (1) scope and purpose (overall aim of the guideline; specific health questions and target population), (2) interested parties involvement (developed by the appropriate stakeholders; consistent with the views of its intended users), (3) rigorously developed (process used to gather and synthesise the evidence; methods to formulate and update the recommendations), (4) clarity and presentation (language; structure and format), (5) applicability (barriers and facilitators to implementation;

**Table 2** Preliminary list of clinical practice guidelines repositories and trauma professional associations

| | |
|---|---|
| 1-Agency for Healthcare Research and Quality | 32-Guideline Central |
| 2-Accreditation Canada | 33-Guidelines International Network |
| 3-Australasian Association for Quality in Healthcare | 34-Headway—the brain injury association |
| 4-American Academy of Orthopaedic Surgeons | 35-International Association for Trauma Surgery and Intensive Care |
| 5-American Association for the Surgery on Trauma | 36-International Society of Orthopaedic and Traumatology |
| 6-American Association of Neurological Surgeons | 37-Italian Society of Orthopaedics and Traumatology |
| 7-American Board of Orthopaedic Surgery | 38-National Association for Healthcare Quality |
| 8-American College of Surgeons | 39-National Guidelines Clearinghouse |
| 9-American Orthopaedic Association | 40-National Institute of Health and Care Excellence |
| 10-American Spinal Injury Association | 41-National Quality Forum |
| 11-American Trauma Society | 42-Norwegian Neurosurgical Association |
| 12-Australian Society of Orthopeadic Surgeons | 43-Ontario Neurotrauma Foundation |
| 13-Australian and New Zealand Society | 44-Orthopaedic Trauma Association |
| 14-Belgian Society for Orthopaedics and Traumatology | 45-Scandinavian Neurotrauma Committee |
| 15-Brain Injury Association of America | 46-Scottish Intercollegiate Guidelines Network |
| 16-Brain Trauma Foundation | 47-Société Française de Chirurgie Orthopédique et Traumatologique |
| 17-British Orthopaedic Association | 48-Society of Critical Care Medicine Trauma Network |
| 18-British Trauma Society | 49-Société Royale Belge de Chirurgie Orthopédique et de Traumatologie |
| 19-Canada's Drug and Health Technology Agency | 50-Spanish Society of Orthopaedic Surgery and Traumatology |
| 20-Centres for Disease Control and Prevention | 51-Society of Trauma Nurses |
| 21-Congress of Neurological Surgeons | 52-Spinal Cord Injury Research Evidence |
| 22-Consortium of Spinal Cord Medicine | 53-Substance Abuse and Mental Health Services Administrations |
| 23-Danish Neurosurgical Society | 54-Swedish Neurosurgical Societ |
| 24-Dutch Institute for Healthcare Improvement | 55-Trauma Association of Canada |

Continued

**Table 2** Continued

| | |
|---|---|
| 25-Eastern Association for the Surgery of Trauma | 56-Trauma Audit Research Network |
| 26-European Association of Neurosurgical Societies | 57-Trauma.org |
| 27-European Federations of National Associations of Orthopaedic and Traumatology | 58-US Department of Health and Human Services |
| 28-European Society for Trauma and Emergency Surgery | 59-US Department of Veterans Affairs |
| 29-Finnish Association of Neuroscience Nurses | 60-Western Trauma Association |
| 30-Finnish Neurosurgical Society | 61-World Health Organisation |
| 31-Greek Society of Orthopaedic and Traumatological Surgery | |

strategies to improve uptake such as integration into a national trauma performance verification process, tools for use at an organisational level, billing guidance for mental health practices, links to other relevant guidelines and tools and demand on healthcare resources) and (6) editorial independence (competing interests reported and addressed). In accordance with guidelines on the SR of CPGs[33] and based on previous SRs of CPGs,[47–49] each domain with a score≥60% will be considered effectively addressed. CPGs will be considered *high quality* if they score≥60% in at least three of the six AGREE II domains, including domain 3 (rigour of development). If three domains or more score ≥60%, and domain 3 score<60%, the CPG will be considered *moderate quality*. CPGs scoring<60% in two domains or more and scoring<50% in domain 3 will be considered *low quality*.

### Data synthesis

CPG recommendations will be classified into the following key areas, as determined by the advisory committee: (1) mental health promotion/illness prevention, (2) screening for mental disorders, (3) assessment, (4) interventions, (5) referral for patients experiencing mental disorders for follow-up and support services and (6) patient and family centered-care approach. We will synthesise the evidence on recommendations for CPGs rated as moderate or high quality (AGREE II)[46] using a recommendation matrix based on the Grading of Recommendations Assessment, Development and Evaluation (GRADE) quality of evidence (high, moderate, low and very low),[50] recommendation strength (low and high),[47–49] health and social determinants (ie, care equity), and whether or not recommendations were formulated using a population-based approach.[45] Matrix data will be extracted independently by pairs or reviewers for each recommendation. We will group the same or very similar recommendations published by more than one CPG. For

CPGs that used a rating system other than GRADE, we will match reported categories to GRADE categories.

For selection, extraction, quality assessment and meta-synthesis, all reviewers will systematically and iteratively pilot test the process on a random selection of articles (200 citations for selection and five CPGs for extraction and quality evaluation and meta-synthesis in each iteration) until acceptable agreement is achieved (kappa>0.8).[32] Any discrepancies during the review process will be resolved by first consulting a senior member of the research team (WP), and then by reaching a consensus among the members of the advisory committee, if necessary.

## Limitation of study

For feasibility reasons, our search strategy was not developed to systematically identify CPGs that do not specifically target mental disorders in trauma populations. Therefore, we may miss recommendations on mental health if they are included in CPGs that target trauma populations in general if no keywords relating to mental health and injury are present in the title or abstract. However, these recommendations are likely to be identified by consulting CPGs repositories and trauma association websites (table 2), which were recognised as the best sources to find relevant guidelines.[51]

## POTENTIAL IMPACT

Recent data on mental health issues in trauma populations are now prompting recognised authorities[23–26] to prioritise this area to further improve the outcomes in injured patients. However, to date, there has been no synthesis of recommendations for improving mental health practices in the trauma field. This SR of CPGs will allow to identify recommendations with the greatest support for implementation, thus taking into account the staff shortage and time constraints omnipresent in healthcare systems. Results may be used by HCPs, trauma programme leaders, hospital administrators and policy-makers to inform local quality improvement initiatives, identify system-wide problems and plan future actions to improve mental health practices.

## ETHICS AND DISSEMINATION

Research ethics approval is not required, as we will conduct secondary analysis of published data. The results of our study will be disseminated in a peer-reviewed journal, at international and national scientific meetings, and an accessible summary will be distributed to stakeholders through professional, healthcare quality and persons with lived experience associations.

## Author affiliations

[1]Population Health and Optimal Practices Research Unit Research Unit (Trauma–Emergency–Critical Care Medicine), CHU de Québec-Université Laval Research Centre, Québec, Quebec, Canada
[2]Faculty of Nursing, Université Laval, Québec City, Quebec, Canada
[3]University of Alberta Hospital, Edmonton, Alberta, Canada
[4]Department of Psychiatry, The University of Melbourne Faculty of Medicine Dentistry and Health Sciences, Melbourne, Victoria, Australia
[5]Department of Critical Care Medicine, University of Calgary, Calgary, Alberta, Canada
[6]Vancouver General Hospital, Vancouver, British Columbia, Canada
[7]University of Sherbrooke, Longueil, Quebec, Canada
[8]Department of Social and Preventive Medicine, Université Laval, Québec City, Quebec, Canada
[9]Institut universitaire en santé mentale Douglas, Montréal, Quebec, Canada
[10]Department of Family Medicine and Emergency Medicine, Université Laval, Québec City, Quebec, Canada
[11]Faculty of Public Health, Erasmus MC, Rotterdam, The Netherlands
[12]Faculté des sciences infirmières, Université de Montréal, Montréal, Quebec, Canada
[13]Centre de recherche de l'Institut universitaire en santé mentale de Montréal, Montréal, Quebec, Canada
[14]Department of Psychiatry, The University of British Columbia, Vancouver, British Columbia, Canada

**Contributors** MB and WP led the development of the protocol and drafted the manuscript. They also secured funding and acts as guarantor for the review. NB, MO and HTS drafted parts of the research methods, validated the search strategy and critically appraised and approved the final manuscript. NG and H-MV assisted with the formulation of the background, objectives and inclusion criteria, and critically appraised and approved the final manuscript. VT, MP, MM, LA, JH, HP, CG, AL and KA contributed to the development of research objectives and concept definitions, validated the search strategy, critically revised and approved the final version of the manuscript. M-AG and LB contributed to the formulation of research objectives and keywords, developed and tested the search strategy and revised and approved the final version of the manuscript. MO, MP, LA, HP, CG and WP acted as an expert in mental health and mental disorders and provided advice on how to integrate the concepts in the protocol.

**Funding** This work is supported by the Canadian Institutes for Health Research (#505680), Fonds de Recherche en Santé du Québec and the Strategy for Patient-Oriented Research-Quebec and the Quebec Nursing Intervention Network. Funders had no role in developing the protocol.

**Competing interests** None declared.

**Patient and public involvement** Patients and/or the public were involved in the design, or conduct, or reporting or dissemination plans of this research. Refer to the Methods section for further details.

**Patient consent for publication** Not applicable.

**Ethics approval** Not applicable.

**Provenance and peer review** Not commissioned; externally peer reviewed.

**ORCID iDs**
Mélanie Bérubé http://orcid.org/0000-0002-6657-3915
Alexandra Lapierre http://orcid.org/0000-0002-8704-4940

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
