## [Reviewer comments · BMJ Open]

ARTICLE DETAILS

TITLE (PROVISIONAL)	Clinical practice guideline recommendations to improve the mental health of adult trauma patients: protocol for a systematic review
AUTHORS	Berube, Melanie; Bradley, Nori; O'Donnell, Meaghan; Stelfox, Henry; Garraway, Naisan; Vasiliadis, Helen-Maria; Turcotte, Valerie; Perreault, M; Menear, Matthew; Archambault, Leonie; Haagsma, Juanita; Provencher, Helene; Genest, Christine; Gagnon, Marc-Aurèle; Bourque, Laurence; Lapierre, Alexandra; Amal, Kalfi; Panenka, William

VERSION 1 – REVIEW

REVIEWER	Daniela de Melo Universidade Federal de Sao Paulo
REVIEW RETURNED	09-Oct-2023

GENERAL COMMENTS	Suggestions for adjustments: Page 8, line 4: "GPCs" Page 8, line 13: "GPCs"
---

REVIEWER	Douglas Zatzick University of Washington
REVIEW RETURNED	12-Jan-2024

GENERAL COMMENTS	The paper addresses an important topic. Clinical Practice Guidelines developed with appropriate theoretical perspectives and applied methods have the potential to enhance real world implementation of optimal trauma focused screening and intervention procedures. A strength of the manuscript is the detailed listing of guidelines to be considered (Table 2). A stated goal of the manuscript is to identify clinical practice guidelines with the greatest support for more widespread implementation with trauma populations (Page 8, Line 26-30). However, a series of key theoretical and applied points not currently addressed in the manuscript markedly reduces confidence that the proposed review will achieve the goal of influencing optimal intervention delivery/implementation across diverse trauma care systems. To begin, the manuscript cites an example from a recent US trauma care system study to explain, at least in part, clinical practice guidelines adherence rates of less than 30% (Citation 27 Bulger et al., 2022). Unfortunately, the manuscript misses the key contextual observation that the survey was conducted between 2019 and 2021, while The American College of Surgeons Committee on Trauma (ACS/COT) requirement for mental health screening and referral and
---

associated clinical practice guidelines were released in 2022. Surveys after 2022 may capture improvements associated with the ACS/COT requirement, so it would be erroneous to assume that pre-requirement surveys could document challenges with clinical practice guideline adherence related to the resource guide requirement.

This example highlights a major challenge for the manuscript, specifically, that different trauma care systems may be better or less prepared to implement best practices identified in clinical guideline recommendations. (See Bulger and Colleagues Catalyzing the Translation of Patient-Centered Research Into United States Trauma Care Systems Medical Care August 2021). For example, in the United States, the ACS/COT has the capacity to require mental health screening and referral procedures and has linked this requirement to every 3 year trauma center verification site visits (see reference 37 cited in the manuscript, Resources for Optimal Care of the Injured Patient 2022). Of the guidelines listed in the manuscript Table 2, The Committee on Trauma Best Practices Guidelines: Screening and Intervention for Substance Use in the Acute Trauma Patients (reference 21, 2022) is unique in that it was developed to inform / support the College screening and referral requirement. Careful review of this Committee on Trauma Guideline reveals a series of domains that will facilitate real world implementation that are not captured in the currently proposed manuscript data synthesis; these domains include, detailed recommendations regarding organizational development for trauma center mental health screening and referral, fastidiously describe Current Procedural Terminology (CPT) billing guidances for mental health screening and referral procedures, and linkage to other American College of Surgeons Committee on Trauma related college mandated procedures and guidelines (i.e., mandated alcohol screening and intervention guidelines).

Also, in order to enhance the ultimate real world applicability of early trauma focused interventions, recent systematic reviews/metanalyses of early interventions for PTSD and related comorbidity after injury have moved beyond limited assessments of intervention treatment effects to incorporate estimates of intervention breadth of applicability and population impact (See Guimmarra and colleagues Early Psychological Interventions for Posttraumatic Stress, Depression, and Anxiety After Traumatic Injury: A Systematic Review and Meta-analysis, Clinical Psychology Review 2018). Preventive intervention researchers have introduced the construct of population impact to describe the amalgamation of treatment effects and breadth of applicability of an intervention procedure (see Koepsell and colleagues. Estimating the population impact of preventive interventions from randomized trials. Am J Prev Med. 2011 Feb;40(2):191-8.). Methodologically, in order to enhance the goal of informing wider trauma care system guideline uptake, much more could be done in the manuscript to describe how the incorporation of the reach/population impact construct as a focus in individual Clinical Practice Guidelines.

The multinational literature review could also be improved. For example, important multisite papers regarding PTSD frequencies in Australian (See O'Donnell and colleagues 2010 American Journal of Psychiatry) and American (See Nathens and Colleagues Psychological Medicine 2007) trauma care systems are not included in the current manuscript.

VERSION 1 – AUTHOR RESPONSE

Reviewer 1 comments	Responses
Page 8, line 4: "GPCs" Page 8, line 13: "GPCs"	We made the modification as suggested.
Reviewer 2 comments	Responses
A stated goal of the manuscript is to identify clinical practice guidelines with the greatest support for more widespread implementation with trauma populations (Page 8, Line 26-30). However, a series of key theoretical and applied points not currently addressed in the manuscript markedly reduces confidence that the proposed review will achieve the goal of influencing optimal intervention delivery/implementation across diverse trauma care systems. To begin, the manuscript cites an example from a recent US trauma care system study to explain, at least in part, clinical practice guidelines adherence rates of less than 30% (Citation 27 Bulger et al., 2022). Unfortunately, the manuscript misses the key contextual observation that the survey was conducted between 2019 and 2021, while The American College of Surgeons Committee on Trauma (ACS/COT) requirement for mental health screening and referral and associated clinical practice guidelines were released in 2022. Surveys after 2022 may capture improvements associated with the ACS/COT requirement, so it would be erroneous to assume that pre-requirement surveys could document challenges with clinical practice guideline adherence related to the resource guide requirement.	We added in the introduction, just before the method, that implementation of the recommendations may have increased after the ACS guidelines were published in December 2022. We also explained that, while these guidelines may have led to an improvement in mental health practices, these practices remain complex and put pressure on the limited resources present in healthcare systems. This justifies the need for a systematic review of the guidelines' recommendations to identify those with the greatest support for implementation, in order to help professionals and organizations in multiple countries, to better target priorities for action.
This example highlights a major challenge for the manuscript, specifically, that different trauma care systems may be better or less prepared to implement best practices identified in clinical guideline recommendations. (See Bulger and Colleagues Catalyzing the Translation of Patient-Centered Research Into United States Trauma Care Systems Medical Care August 2021). For example, in the United States, the ACS/COT has the capacity to require mental health screening and referral procedures and has linked this requirement to every 3 year trauma center verification site visits (see reference 37 cited in the manuscript, Resources for Optimal Care of	Strategies facilitating the applicability of recommendations by organizations is one of the criteria used to assess the quality of guidelines using the Appraisal of Guidelines Research and Evaluation (AGREE) II grid. ACS guidelines will consequently score very highly on this criterion. We have added examples to the "applicability" evaluation criterion in the "Quality section" in order to make it more explicit and illustrate how tools provided with guidelines can optimize the uptake of recommendations in trauma centers.

the Injured Patient 2022). Of the guidelines listed in the manuscript Table 2, The Committee on Trauma Best Practices Guidelines: Screening and Intervention for Substance Use in the Acute Trauma Patients (reference 21, 2022) is unique in that it was developed to inform / support the College screening and referral requirement. Careful review of this Committee on Trauma Guideline reveals a series of domains that will facilitate real world implementation that are not captured in the currently proposed manuscript data synthesis; these domains include, detailed recommendations regarding organizational development for trauma center mental health screening and referral, fastidiously describe Current Procedural Terminology (CPT) billing guidances for mental health screening and referral procedures, and linkage to other American College of Surgeons Committee on Trauma related college mandated procedures and guidelines (i.e., mandated alcohol screening and intervention guidelines).	
Also, in order to enhance the ultimate real world applicability of early trauma focused interventions, recent systematic reviews/metanalyses of early interventions for PTSD and related comorbidity after injury have moved beyond limited assessments of intervention treatment effects to incorporate estimates of intervention breadth of applicability and population impact (See Guimmarra and colleagues Early Psychological Interventions for Posttraumatic Stress, Depression, and Anxiety After Traumatic Injury: A Systematic Review and Meta-analysis, Clinical Psychology Review 2018). Preventive intervention researchers have introduced the construct of population impact to describe the amalgamation of treatment effects and breadth of applicability of an intervention procedure (see Koepsell and colleagues. Estimating the population impact of preventive interventions from randomized trials. Am J Prev Med. 2011 Feb;40(2):191-8.). Methodologically, in order to enhance the goal of informing wider trauma care system guideline uptake, much more could be done in the manuscript to describe how the incorporation of the reach/population impact construct as a focus in individual Clinical Practice Guidelines.	The population-based approach is indeed important in a mental health context to ensure that access to care is maintained while meeting patients' needs. So, to take this into account, we will, as described in the “CPGs selection and data extraction section”, extract data to determine whether recommendations are formulated to take this approach into account. Also, as explained in the “Data synthesis section”, we will add to our data synthesis matrix whether the recommendations were formulated according to a population-based approach or not. We have also added the suggested reference to support the importance of considering this approach.
The multinational literature review could also be improved. For example, important multisite	We added these references in the 2nd paragraph of the “Introduction section”, and we added that

papers regarding PTSD frequencies in Australian (See O'Donnell and colleagues 2010 American Journal of Psychiatry) and American (See Nathens and Colleagues Psychological Medicine 2007) trauma care systems are not included in the current manuscript.	the statistics describing mental disorders apply to many high-income countries.
--	---

VERSION 2 – REVIEW

REVIEWER	Douglas Zatzick University of Washington
REVIEW RETURNED	04-Mar-2024

GENERAL COMMENTS	Overall, the authors have responded to the reviewer critique. One minor note, it was not clear where the Giummarra et al manuscript had been added to the paper / references (the cover letter states the reference is to be added).
--

VERSION 2 – AUTHOR RESPONSE

Response to reviewer

Comments to the Author:

Overall, the authors have responded to the reviewer critique. One minor note, it was not clear where the Giummarra et al manuscript had been added to the paper / references (the cover letter states the reference is to be added).

Response:

We added the Giummarra reference as suggested (reference no 45).